# MODEL AGNOSTIC META-LEARNING ON TREES

## ABSTRACT

In meta-learning, the knowledge learned from previous tasks is transferred to new ones, but this transfer only works if tasks are related, and sharing information between unrelated tasks might hurt performance. A fruitful approach is to share gradients across similar tasks during training, and recent work suggests that the gradients themselves can be used as a measure of task similarity. We study the case in which datasets associated to different tasks have a hierarchical, tree structure. While a few methods have been proposed for hierarchical meta-learning in the past, we propose the first algorithm that is model-agnostic, a simple extension of MAML. As in MAML, our algorithm adapts the model to each task with a few gradient steps, but the adaptation follows the tree structure: in each step, gradients are pooled across task clusters, and subsequent steps follow down the tree. We test the algorithm on linear and non-linear regression on synthetic data, and show that the algorithm significantly improves over MAML. Interestingly, the algorithm performs best when it does not know in advance the tree structure of the data.

## 1 INTRODUCTION

Deep learning models require a large amount of data in order to perform well when trained from scratch. When data is scarce for a given task, we can *transfer* the knowledge gained in a source task to quickly learn a target task, if the two tasks are related. The field of *Multi-task learning* studies how to learn multiple tasks simultaneously, with a single model, by taking advantage of task relationships (Ruder (2017), Zhang & Yang (2018)). However, in Multi-task learning models, a set of tasks is fixed in advance, and they do not generalize to new tasks. The field of of *Meta-learning* is inspired by the ability of humans to learn how to quickly learn new tasks, by using the knowledge of previously learned ones.

Meta-learning has seen a widespread use in multiple domains, especially in recent years and after the advent of Deep Learning (Hospedales et al. (2020)). However, there is still a lack of methods for sharing information across tasks in meta-learning models, and the goal of our work is to fill this gap. In particular, a successful model for meta-learning, MAML (Finn et al. (2017)), does not diversify task relationships according to their similarity, and it is unclear how to modify it for that purpose.

In this work, we contribute the following:

- We propose a novel modification of MAML to account for a hierarchy of tasks. The algorithm uses the tree structure of data during adaptation, by pooling gradients across tasks at each adaptation step, and subsequent steps follow down the tree (see Figure 1a).
- We introduce new benchmarks for testing a hierarchy of tasks in meta-learning on a variety of synthetic non-linear (sinusoidal) and multidimensional linear regression tasks.
- We compare our algorithm to MAML and a baseline model, where we train on all tasks but without any meta-learning algorithm applied. We show that the algorithm has a better performance with respect to both of these models in the sinusoidal regression task and the newly introduced synthetic task because it exploits the hierarchical structure of the data.

## 2 RELATED WORK

The problem of quantifying and exploiting task relationships has a long history in Multi-task learning, and is usually approached by parameter sharing, see Ruder (2017), Zhang & Yang (2018) for

reviews. However, Multi-task Learning is fundamentally different from Meta-learning as it does not consider the problem of generalizing to new tasks (Hospedales et al. (2020)). Recent work includes Zamir et al. (2018), who studies a large number of computer vision tasks and quantifies the transfer between all pairs of tasks. Achille et al. (2019) proposes a novel measure of task representation, by assigning an importance score to each model parameter in each task. The score is based on the gradients of each task's loss function with respect to each model parameter. This work suggests that gradients can be used as a measure of task similarity, and we use this insight in our proposed algorithm.

In the context of Meta-learning, a few papers have been published on the problem of learning and using task relationships in the past months. The model of Yao et al. (2019) applies hierarchical clustering to task representations learned by an autoencoder, and uses those clusters to adapt the parameters to each task. The model of Liu et al. (2019) maps the classes of each task into the edges of a graph, it meta-learns relationships between classes and how to allocate new classes by using a graph neural network with attention. However, these algorithms are not model-agnostic, they have a fixed backbone and loss function, and are thus difficult to apply to new problems. Instead, we design our algorithm as a simple generalization of Model-agnostic meta-learning (MAML, Finn et al. (2017)), and it can be applied to any loss function and backbone.

A couple of studies looked into modifying MAML to account for task similarities. The work of Jerfel et al. (2019) finds a different initial condition for each cluster of tasks, and applies the algorithm to the problem of continual learning. The work of Katoch et al. (2020) defines parameter updates for a task by aggregating gradients from other tasks according to their similarity. However, in contrast with our algorithm, both of these models are not hierarchical, tasks are clustered on one level only and cannot be represented by a tree structure. As far as we know, ours is the first model-agnostic algorithm for meta-learning that can be applied to a tree structure of tasks.

## 3 THE META-LEARNING PROBLEM

We follow the notation of Hospedales et al. (2020). We assume the existence of a distribution over tasks $\tau$ and, for each task, a distribution over data points $\mathcal{D}$ and a loss function $\mathcal{L}$. The loss function of the meta-learning problem, $\mathcal{L}^{meta}$, is defined as an average across both distributions of tasks and data points:

$$\mathcal{L}^{meta}\left(\boldsymbol{\omega}\right) = \mathop{\mathbb{E}}_{\tau} \mathop{\mathbb{E}}_{\mathcal{D}|\tau} \mathcal{L}_{\tau}\left(\boldsymbol{\theta}_{\tau}(\boldsymbol{\omega}); \mathcal{D}\right) \tag{1}$$

The goal of meta-learning is to minimize the loss function with respect to a vector of meta-parameters $\boldsymbol{\omega}$. The vector of parameters $\boldsymbol{\theta}$ is task-specific and depends on the meta-parameters $\boldsymbol{\omega}$. Different meta-learning algorithms correspond to a different choice of $\boldsymbol{\theta}_{\tau}(\boldsymbol{\omega})$. We describe below the choice of TreeMAML, the algorithm proposed in this study.

During meta-training, the loss is evaluated on a sample of $m$ tasks, and a sample of $n_v$ validation data points for each task

$$\mathcal{L}^{meta}\left(\boldsymbol{\omega}\right) = \frac{1}{mn_v} \sum_{i=1}^{m} \sum_{j=1}^{n_v} \mathcal{L}_{\tau_i}\left(\boldsymbol{\theta}_{\tau_i}(\boldsymbol{\omega}); \mathcal{D}_{ij}\right) \tag{2}$$

For each task $i$, the parameters $\boldsymbol{\theta}_{\tau_i}$ are learned by a set of $n_t$ training data points, distinct from the validation data. During meta-testing, a new (target) task is given and the parameters $\boldsymbol{\theta}$ are learned by a set of $n_r$ target data points. In this work, we also use a batch of training data points to adapt $\boldsymbol{\theta}$ at test time. No training data is used to compute the final performance of the model, which is computed on separate test data of the target task.

### 3.1 MAML

MAML aims at finding the optimal initial condition $\omega$ from which a good parameter set can be found, separately for each task, after $K$ gradient steps (Finn et al. (2017)). For task $i$, we define the single gradient step with learning rate $\alpha$ as

$$U_i(\boldsymbol{\omega}) = \boldsymbol{\omega} - \frac{\alpha}{n_t} \sum_{j=1}^{n_t} \nabla\mathcal{L}(\boldsymbol{\omega}; \mathcal{D}_{ij}) \tag{3}$$

Then, MAML with $K$ gradient steps corresponds to $K$ iterations of this step (here we assume that the same batch of training data points is used at each step, because these are task specific)

$$\boldsymbol{\theta}_{\tau_i}(\boldsymbol{\omega}) = U_i(U_i(...U_i(\boldsymbol{\omega}))) \qquad (K \text{ times}) \qquad (4)$$

This update is usually referred to as *inner loop*, and is performed separately for each task, while optimization of the loss 2 is referred to as *outer loop*.

## 3.2 TreeMAML

We propose to modify MAML in order to account for a hierarchical structure of tasks. The idea is illustrated in Figure 1.

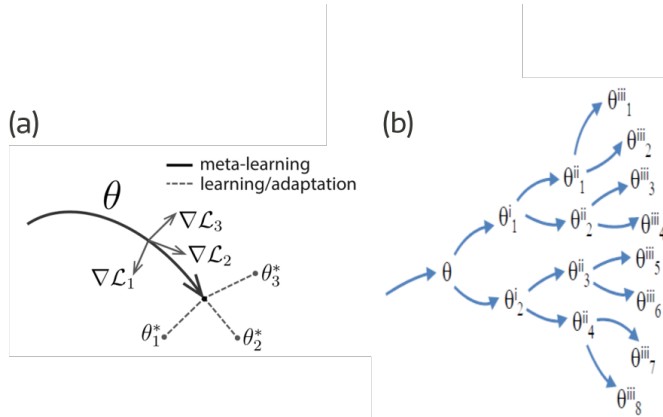

Figure 1: Illustration of the MAML(a), and TreeMAML(b) algorithms. Both algorithms are designed to quickly adapt to new tasks with a small number of training samples. MAML achieves this by introducing a gradient step in the direction of the single task. TreeMAML follows a similar approach but exploiting the relationship between tasks by introducing a hierarchical aggregation of the gradients.

At each gradient step $k$, we assume that tasks are aggregated into $C_k$ clusters, and the parameters for each task are updated according to the average gradient across tasks within the corresponding cluster (in Fig.1b, we use $K = 3$ steps and $C_1 = 2$, $C_2 = 4$, $C_3 = 8$). We denote by $\mathcal{T}_c$ the set of tasks in cluster $c$. Then, the gradient update for the parameters of each task belonging to cluster $c$ is equal to

$$U_c(\boldsymbol{\omega}) = \boldsymbol{\omega} - \frac{\alpha}{n_t |\mathcal{T}_c|} \sum_{i \in \mathcal{T}_c} \sum_{j=1}^{n_t} \nabla \mathcal{L}(\boldsymbol{\omega}; \mathcal{D}_j^{(i)}) \qquad (5)$$

Furthermore, we denote by $c_i^k$ the cluster to which task $i$ belongs at step $k$. Then, TreeMAML with $k$ gradient steps corresponds to $K$ iterations of this step

$$\boldsymbol{\theta}_{\tau_i}(\boldsymbol{\omega}) = U_{c_i^K}(U_{c_i^{K-1}}(...U_{c_i^1}(\boldsymbol{\omega}))) \qquad (6)$$

The intuition is the following: if each task has scarce data, gradient updates for single tasks are noisy, and adding up gradients across similar tasks increases the signal. Note that we recover MAML if $C_k$ is equal to the total number of tasks $m$ at all steps. On the other hand, if $C_k = 1$ then the inner loop would take a step with a gradient averaged across all tasks.

Because at one specific step the weight updates are equal for all tasks within a cluster, it is possible to define the steps of the inner loop update per cluster $c$ instead of per task $\boldsymbol{\theta}_{\tau_i}$. Given a cluster $c$ and its parent cluster $p_c$ in the tree, the update at step $k$ is given by

$$\boldsymbol{\theta}_{c,k} = \boldsymbol{\theta}_{p_c,k-1} - \frac{\alpha}{n_t |\mathcal{T}_c|} \sum_{i \in \mathcal{T}_c} \sum_{j=1}^{n_t} \nabla \mathcal{L}(\boldsymbol{\theta}_{p_c,k-1}; \mathcal{D}_{ij}) \qquad (7)$$

where $\boldsymbol{\theta}_k^c$ is the parameter value for cluster $c$ at step $k$. In terms of the notation used in expression 6, we have the equivalence $\boldsymbol{\theta}_{\tau_i}(\boldsymbol{\omega}) = \boldsymbol{\theta}_{c_i,K}$, which depends on the initial condition $\boldsymbol{\omega}$. The full procedure is described in Algorithm 1

We consider two versions of the algorithm, depending on how we obtain the tree structure (similar to Srivastava & Salakhutdinov (2013)):

- *Fixed tree*. The tree is fixed by the knowledge of the tree structure of tasks, when this structure is available. In that case, the values of $C_k$ are determined by such tree.

- *Learned tree*. The tree is unknown *a priori*, and is learned using a hierarchical clustering algorithm. In that case, the values of $C_k$ are determined at each step as a result of the clustering algorithm.

In the latter case, we cluster tasks based on the gradients of each task loss, consistent with recent work (Achille et al. (2019)). After each step $k$ at cluster $c_i$, the clustering algorithm takes as input the gradient vectors of the children tasks $i$

$$\mathbf{g}_{ik} = \frac{1}{n_t} \sum_{j=1}^{n_t} \nabla \mathcal{L}(\boldsymbol{\theta}_{c_i,k}; \mathcal{D}_{ij}) \tag{8}$$

and these gradients are further allocated into clusters according to their similarity. The clustering algorithm is described in subsection 3.3.

Similar to MAML, adaptation to a new task is performed by computing $\boldsymbol{\theta}^{(i)}(\boldsymbol{\omega})$ on a batch of data of the target task. In order to exploit task relationships, we first reconstruct the tree structure by using a batch of training data and then we introduce the new task.

---

**Algorithm 1** TreeMAML

---

**Require:** distribution over tasks $p(\tau)$; distribution over data for each task $p(\mathcal{D}|\tau)$;
**Require:** number of inner steps $K$; number of training tasks $m$; learning rates $\alpha, \beta$;
**Require:** number of clusters $C_k$ for each step $k$; loss function $\mathcal{L}_\tau(\boldsymbol{\omega}, \mathcal{D})$ for each task
    randomly initialize $\boldsymbol{\omega}$
    **while** not done **do**
        sample batch of $i = 1 : m$ tasks $\{\tau_i\} \sim p(\tau)$
        for all tasks $i = 1 : m$ initialize a single cluster $c_i = 1$
        initialize $\boldsymbol{\theta}_{1,0} = \boldsymbol{\omega}$
        **for** steps $k = 1 : K$ **do**
            **for** tasks $i = 1 : m$ **do**
                sample batch of $j = 1 : n_v$ data points $\{\mathcal{D}_{ij}\} \sim p(\mathcal{D}|\tau_i)$
                evaluate gradient $\mathbf{g}_{ik} = \frac{1}{n_t} \sum_{j=1}^{n_t} \nabla \mathcal{L}_{\tau_i}(\boldsymbol{\theta}_{c_i,k-1}; \mathcal{D}_{ij})$
            **end for**
            regroup tasks into $C_k$ clusters $\mathcal{T}_c = \{i : c_i = c\}$
            according to similarity of $\{\mathbf{g}_{ik}\}$ and parent clusters $\{p_c\}$
            update $\theta_{c,k} = \theta_{p_c,k-1} - \frac{\alpha}{|\mathcal{T}_c|} \sum_{i \in \mathcal{T}_c} \mathbf{g}_{ik}$ for all clusters $c = 1 : C_k$
        **end for**
        update $\boldsymbol{\omega} \leftarrow \boldsymbol{\omega} - \beta \frac{1}{mn_v} \sum_{i=1}^{m} \sum_{j=1}^{n_v} \nabla_{\boldsymbol{\omega}} \mathcal{L}_{\tau_i}(\boldsymbol{\theta}_{c_i,K}(\boldsymbol{\omega}); \mathcal{D}_{ij})$
    **end while**

---

### 3.3 Clustering Algorithm

In the learned tree case we employ a hierarchical clustering algorithm to cluster the gradients of our model parameters. We specifically opt for an online clustering algorithm to maximise computational efficiency at test time and scalability. When a new task is evaluated, we reuse the tree structure that was generated for a training batch and add the new task. This saves us from computing a new task hierarchy from scratch for every new task. Moreover, with offline hierarchical clustering all the data needs to be available to the clustering algorithm at the same time, which becomes a problem when dealing with larger batch sizes. Therefore online clustering favours scalability.

We follow the online top down (OTD) approach set out by Menon et al. (2019) and adapt this to approximate to non-binary tree structures. Our clustering algorithm is shown in Algorithm 2. Specifically, we introduce two modifications to the original OTD algorithm:

- **Maximum Tree Depth Parameter** $D$: This is equivalent to the number of inner steps to take in the TreeMAML, since the tree is a representation of the inner loop where each layer in the tree represents a single inner step.

- **Non-binary Tree Approximation**: We introduce a hyperparameter $\xi$ which represents how far the similarity of a new task needs to be to the average cluster similarity in order to be considered child of a that same cluster. This is not an absolute value of distance, but it is a multiplicative factor of the standard deviation of the intracluster similarities. Introducing this factor allows clusters at any level to have a number of children greater than two.

---

**Algorithm 2** Online top down (OTD) - Non-binary

---

**Require:** origin cluster node $C$ with a given set of children $A = \{x_1, x_2, ..x_N\}$
**Require:** new task $x$; maximum depth allowed $D$; similarity metric, $\omega()$
**Require:** standard deviation multiplicative hyperparameter $\xi$;
  **if** $|A| = 0$ **then**
    new task becomes a new child $A = \{x\}$
  **else if** $|A| = 1$ **then**
    add new task to set of children $A \leftarrow A \cup \{x\}$
  **else if** $\omega(A \cup \{x\}) > \omega(A)$ **then**
    identify most similar child $x_* = \arg\min_{x_i}(\omega(\{x_i, x\}))$
    **if** reached maximum depth $C_{\text{depth}} + 1 = D$ **then**
      add new task to set of children $A \leftarrow A \cup \{x\}$
    **else**
      recursively perform OTD to create new node $C' = \text{OTD}(x_*, x)$
      add new node to set of children $A \leftarrow (A \setminus \{x_*\}) \cup C'$
    **end if**
  **else if** $\omega(A \cup \{x\}) < \omega(A) - \xi\sigma_T$ **then**
    current node and new task become children to new cluster $A \leftarrow \{C, x\}$
  **else**
    add new task to set of children $A \leftarrow A \cup \{x\}$
  **end if**

---

## 4 EXPERIMENTS

### 4.1 FIXED TREEMAML

In this section we report experiments where we assumed knowledge about the structure of the underlying tasks and used this specifically to aggregate the gradients.

**Experiment 1: Sinusoidal Regression Tasks** We start with a modification of the regression problem used in the Finn et al. (2017) paper, the regression of a sine wave, where the amplitude and phase of the sinusoid are varied between tasks. We introduce a single modification to the experiment, in the original dataset the amplitude varies within [0.1, 5.0] and the phase varies within [0, $\pi$], but in our experiments we have selected subsets of these values to simulate structured data, as shown in Figure 2. To increase the difficulty of the task, we added different levels of noise to the datasets.

As in the original experiment, the data points for each task are sampled uniformly between [-5.0, 5.0]. During training and testing, the loss used is the mean-squared error. The model is a neural network with 2 hidden layers of size 40 with ReLU nonlinearities. We assume a relationship between tasks as the one represented in Figure 2 (c), where the total depth of the tree is 2 and 4 leaves in total. These leaves represent the finer level clusters and each task is fed to the TreeMAML algorithm with an assigned label corresponding to one of these.

We evaluate performance by fine-tuning the model learned by MAML, TreeMAML and baseline on K = {3, 5, 100} data points. We also evaluate for the case where K = 5 data points were provided for

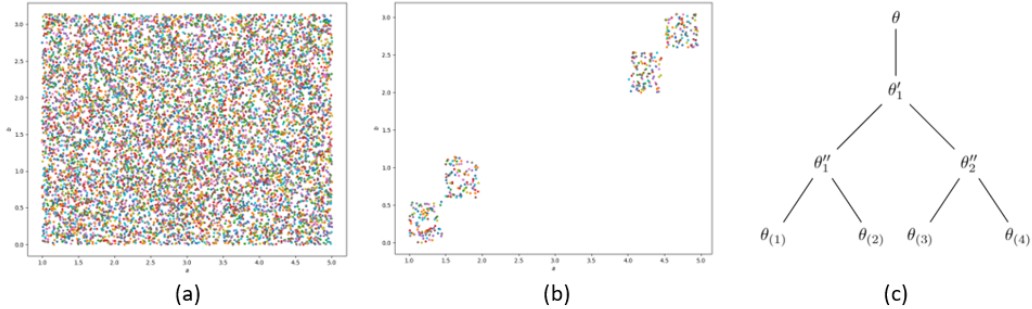

Figure 2: We show the tasks distribution over amplitude and phase of the original dataset used in Finn et al. (2017) (a) and in our modified dataset (b). The change in tasks distribution was introduced to simulate structured data. (c) Shows the equivalent fixed tree structure that was used to represent the data.

training, but a single data point was provided at meta-testing, see point marked as a star in Figure 3.

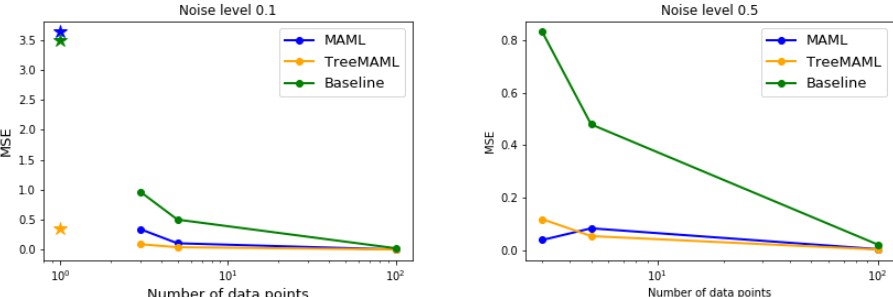

Figure 3: Results of sinusoidal regression task for Fixed TreeMAML, MAML and baseline. The plots show the loss (MSE) at K shot regression for different numbers meta-testing samples. (a) shows the results for noise level 0.1 and (b) shows results for noise level 0.5. Fixed TreeMAML always performs performs better than or as good as MAML and the baseline, especially in the case of a small number of data points being available.

**Experiment 2a: Multidimensional Linear Regression Task** Here we consider a multidimensional linear regression problem $y = \sum_{i=1}^{64} P_i x_i + \eta$ where the tasks are randomly sampled from a set of 4 defined clusters of multidimensional parameters $P$. $\eta$ is randomly some generated Gaussian noise. Even in this case, the parameter clusters are arranged hierarchically such that $C_1 = 2, C_2 = 4$. The data points for the tasks are sampled uniformly $x_i \sim U[-5.0, 5.0]$ for all training and testing tasks where. The models are then trained and tested on a set of tasks with $K = 4, 8, 16, 32, 64$ and $128$ data points.

## 4.2 LEARNED TREEMAML

In this section we report experiments where we assume no prior knowledge of the underlying structure of the data. Therefore the hierarchy of the data is learnt per-batch using the modified OTD algorithm described in section 3.3.

**Experiment 2b: Learned Tree with Multidimensional Linear Regression Tasks** For the multidimensional linear regression task, we set the maximum depth of the clustering algorithm to 2 and we use the cosine similarity metric. This is equivalent to 3 inner steps, beacuse there one last step that is task-specific; therefore for these experiments MAML is also set to perform 3 inner steps.

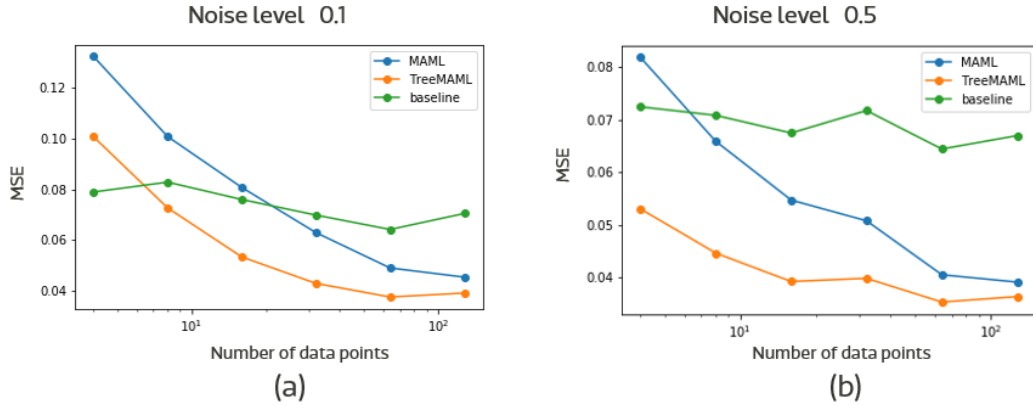

(a)  (b)

Figure 4: Results of the multidimensional (N=64) linear regression task for Fixed TreeMAML, MAML and baseline for varying number of task data points $K = 4, 8, 16, 32, 64$ and 128. Similarly to the sinusoidal regression task, TreeMAML performs better than MAML, especially when the number of tasks data points K is low.

| Model | K=5 | k=10 | k=20 |
|---|---|---|---|
| MAML | $1.025 \pm 0.068$ | $0.950 \pm 0.048$ | $0.785 \pm 0.028$ |
| Baseline | $1.293 \pm 0.074$ | $1.055 \pm 0.047$ | $1.139 \pm 0.061$ |
| **Fixed TreeMAML (ours)** | $\mathbf{0.621 \pm 0.038}$ | $\mathbf{0.535 \pm 0.024}$ | $\mathbf{0.483 \pm 0.016}$ |
| **Learned TreeMAML (ours)** | $0.758 \pm 0.047$ | $\mathbf{0.510 \pm 0.024}$ | $0.495 \pm 0.018$ |

Table 1: Loss (MSE) $\pm 95\%$ confidence intervals on multidimensional linear regression task, averaged over 400 meta-testing tasks. The results are presented for a varying numbers of K data points and a noise level of 0.01

In Table 1 we show that TreeMAML outperforms the Baseline and MAML across all numbers of data points. Learned TreeMAML performs relatively better for a larger number of data points; this is expected because, as the number of data points increases, the gradients used to cluster the tasks will be less affected by the noise and become more accurate, leading also to better clustering.

**Experiment 3: Mixed Synthetic Regression Tasks** Here we follow a similar setup as described in Yao et al. (2019), however we choose different parameters for our tasks in order to have better defined clusters of tasks. Again, we sample data points for the regression tasks as $x \sim U[-5.0, 5.0]$. We define a total of 6 clusters as follow: (1) *Linear - Positive Slopes*: $y = a_{l+}x + b_{l+}$, $a_{l+} \sim U[1.0, 2.0]$ and $b_{l+} \sim U[0, 1.0]$; (2) *Quadratic - Positive Slopes*: $y = a_{q+}x^2 + b_{q+}x + c_{q+}$, $a_{q+} \sim U[0.1, 0.2]$, $b_{q+} \sim U[1.0, 2.0]$ and $c_{q+} \sim U[2.0, 3.0]$; (3) *Linear - Negative Slopes*: $y = a_{l-}x + b_{l-}$, $a_{l-} \sim U[-2.0, -1.0]$ and $b_{l-} \sim U[-1.0, 0]$; (4) *Quadratic - Negative Slopes*: $y = a_{q-}x^2 + b_{q-}x + c_{q-}$, $a_{q-} \sim U[-0.2, -0.1]$, $b_{q-} \sim U[-2.0, -1.0]$ and $c_{q-} \sim U[-3.0, -2.0]$; (5) *Cubic*: $y = a_cx^3 + b_cx^2 + c_cx + d_c$, $a_c \sim U[-0.1, 0.1]$, $b_c \sim U[-0.2, 0]$, $c_c \sim U[-2.0, -1.0]$ and $d_c \sim U[0, 3.0]$; (6) *Sinusoidals*: $y = a_s \sin(x) + b_s$, $a_s \sim U[4.0, 5.0]$ and $b_s \sim U[2.0, \pi]$. To each of these we add a noise variable sampled for each data point sampled from $\sim U[-0.01, 0.01]$. We logically arranged this set of tasks in a tree structure as the one shown in Figure 5 and we used this as reference for the Fixed TreeMAML experiments. We train all tasks with K=10 data points. The model used is a neural network with 2 hidden layers of size 40 with ReLU nonlinearities.

The results stated here are averaged over 600 test tasks (100 from each cluster). For these experiments we set the number of MAML inner steps to 4 and the depth of the Learnt TreeMAML to 3 in order to match the number of steps as in the hierarchy in Figure 5. The results for the models are shown in Table 2 for a number of epochs $E = \{1, 5, 10\}$. The results confirm that TreeMAML has a clear advantage over the MAML and baseline algorithms.

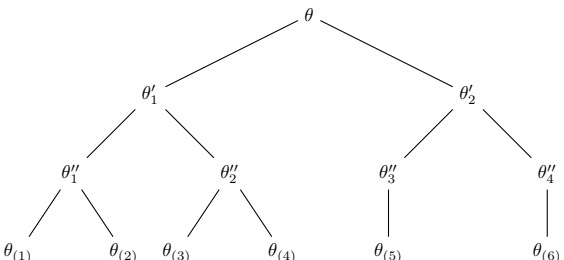

Figure 5: This tree structure shows the task relationship that was logically designed and used for the Fixed TreeMAML.

| Model | 1 epoch | 5 epochs | 10 epochs |
|---|---|---|---|
| MAML | $4.135 \pm 1.148$ | $2.340 \pm 0.632$ | $1.804 \pm 0.290$ |
| Baseline | $20.745 \pm 1.416$ | $20.519 \pm 1.406$ | $20.130 \pm 1.390$ |
| **Fixed TreeMAML (ours)** | $\mathbf{1.955 \pm 0.279}$ | $\mathbf{1.427 \pm 0.219}$ | $\mathbf{1.397 \pm 0.222}$ |
| **Learned TreeMAML (ours)** | $\mathbf{1.637 \pm 0.255}$ | $\mathbf{1.338 \pm 0.225}$ | $\mathbf{1.366 \pm 0.240}$ |

Table 2: Performance (MSE loss) $\pm 95\%$ confidence intervals on mixed regression task, averaged over 600 tasks. N-epoch results on tasks with K=10 data points and a 0.01 noise level.

## 5 DISCUSSION

We proposed a simple modification of MAML to address the problem of meta-learning hierarchical task distributions. The proposed algorithm is based on the intuitive notion that learning tasks by gradient descent may benefit from gradient sharing, across similar tasks. Inspired by recent work Achille et al. (2019), we use the insight that similarity of tasks can be measured by the similarity of gradient themselves, thus reducing the problem of task transfer to gradient clustering.

We show that the new algorithm, which we term TreeMAML, performs better than MAML when the task structure is hierarchical. However, our tests are so far limited to synthetic data, and future work will have to validate our approach to more realistic settings. For example, some computer vision datasets have a hierarchical structure (Deng et al. (2009)) and thus may represent a good test bed for our algorithm.

We presented a very basic instance of our algorithm, thaht can be improved in several ways. For example, not all parameters need to be adapted, and recent work suggests that removing the inner loop for a subset of parameters increases performance, especially when data is scarce (Zintgraf et al. (2019),Raghu et al. (2020)). Another possible modification of the algorithm is to have a different number of clusters for different subset of parameters. In general, given that our algorithm is a relatively simple modification of MAML, several tricks to improve training of the latter could be used for our algorithm as well.

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
