# OpenReview forum: "Model agnostic meta-learning on trees"
_ICLR.cc/2021/Conference — Reject_

### Official Review · AnonReviewer2 · 2020-10-27
**Interesting ideas, but execution is weak.**

**Rating:** 3
**Confidence:** 3

**Review:**

#### Summary
- In the context of gradient-based meta-learning for few-shot learning, the authors propose TreeMAML, an algorithm that leverages the existence of a tree structure in a task distribution in order to pool inner-loop gradients between tasks. More specifically, the inner-loop corresponds to a level-by-level traversal from root to leaves, where at each step, task-leaves that are children to a common node in the current level average their gradients for this step's update.
- The authors consider two cases: one in which the ground-truth tree structure is known (Fixed TreeMAML) and one in which it is unknown and must be discovered online (Learned TreeMAML). The authors extend a hierarchical clustering algorithm from prior work for this purpose.
- The authors evaluate TreeMAML in three scenarios: sinusoid regression (Finn et al. 2017), linear regression, and mixed regression (Yao et al. 2019). TreeMAML compares favorably against MAML and a naive multi-task learning baseline.
- Interestingly, Learned TreeMAML also consistently outperforms Fixed TreeMAML.


#### Strengths
- The authors investigate an important and under-considered problem in meta-learning: how to leverage structure within a task distribution.

- The proposed TreeMAML algorithm is conceptually simple.

#### Weaknesses
- The motivation for the TreeMAML algorithm is very weak. Yes, averaging gradients across tasks might decrease variance, but presumably always increases bias. This crucial trade-off is not even mentioned.

- The fact that Learned outperforms Fixed is troubling. It suggests that TreeMAML is not properly leveraging the ground-truth task hierarchy. Dissecting Learned to look at the tree structure it proposes would help diagnose this issue.

- This is a purely empirical paper which only presents results in toy regression settings. More comprehensive empirical evaluation is needed, e.g. the image classification benchmark proposed in Yao et al. (2019), which is perfectly suitable for this paper (and indeed the authors took Experiment 3 from this work).

- The comparison between TreeMAML and MAML might not be very fair. MAML is artificially constrained to use the same number of inner-loop steps as the depth of the task tree. Since MAML makes no assumptions about the task tree, this should be a tunable hyperparameter.

#### Recommendation
- I currently recommend a clear reject (3). Given the weaknesses outlined above, this submission does not meet the bar for acceptance.

#### Questions
- How is the Baseline model trained? Its MSE in Table 2 is uncommonly high.

#### Minor suggestions
- Please fix the numerous typos throughout the submission. Just in the first paragraph: inappropriate capitalization of multi-task and meta-learning; "The field of of".
- K is used to denote the number of inner gradient steps in Alg. 1, but the number of datapoints per task in Sec. 4.1.
- The num_shots=3 case in Fig. 3b directly contradicts the caption.

---

> ### Author Response · Authors · 2020-11-23
> **Answer to "weaknesses section"**
>
> Thanks for taking the time to review our paper:
>
> - The possible imbalance in bias-variance is an interesting observation. We believe that this algorithm may increase the bias towards the cluster, but the assumption is that the new task belongs to these clusters.
>
> - The fact that the Learned TreeMAML outperform the fixed one is surprising in the synthetic dataset, but inside statistical errors. We have checked the structure of the learned tree and for the simple synthetic examples, the fixed and the learned tree are the same.
>
> - We agree that more comprehensive empirical data is needed and we will work on it for future submissions.
>
> - We used the same numbers of inners steps for MAML than for TreeMAML to be as fair as possible. In the case of MAML the higher that is the number of inner steps the faster that algorithm converges and the higher is the accuracy, as shown in fig.3 of the MAML paper.

---

### Official Review · AnonReviewer4 · 2020-10-27
**Model agnostic meta-learning on trees**

**Rating:** 5
**Confidence:** 3

**Review:**

##########################################################################

Summary:
The paper studies a simple modification of MAML to address the problem of meta-learning hierarchical
task distributions. It is based on the assumption that learning tasks by gradient descent may benefit from gradient sharing, across similar tasks. The authors convert the problem of task transfer to gradient clustering by considering the point that the similarity of tasks can be measured by the similarity of gradients.

##########################################################################

Reasons for score:


Overall, I vote for rejecting this paper. Unfortunately, I found the novelty of the paper very low. I believe that this paper is like an incremental study of exist work like MAML[1] and HSML[2].
[1]https://arxiv.org/abs/1703.03400
[2]http://proceedings.mlr.press/v97/yao19b/yao19b.pdf


##########################################################################
Pros:
- The paper is clear and cites the most relevant research studies and papers.
- The experimental results presents the promising performance of the method.
- It is a strength point of the paper that the authors address both foxed tree and  learned tree structures.

##########################################################################

Cons:
- The novelty of the paper is very low. The authors refer to [2] in the section 2 of the paper and mention that it is not task agnostic.
a) I believe HSML[2] is also built on MAML and not sure about this claim.
- Lack of extensive experiments:
a) The paper does not include any experiment or discussion comparing the proposed method with HSML[2] or other exist method. I would suggest the authors to consider running more experiments.
b) The time complexity of the method has not been discussed.

##########################################################################

Questions during rebuttal period:


Please address and clarify the cons above


#########################################################################

---

> ### Author Response · Authors · 2020-11-23
> **Answer to "Cons"**
>
> Thanks for your precise review. We provide clarifications to the "Cons" below:
>
> 1. Novelty:
> - We believe that the Novelty of the approach is to modify the gradient descent to allow the NN to use the structure present on the data. And that we have shown that we can outperform MAML using this knowledge. This objective is already achieved with a fixed tree. There are many datasets that have an underlying known tree-structure as CIFAR, and Imagenet. And this is the case of some real-world data also.
> - The fact that this paper is using MAML as a starting point does not speak against the novelty of the concept. MAML is the base of many meta-learning papers as REPTILE, foMAML, etc. We believe that our algorithm is different from the one used in [2] since we use the gradients itself for the clustering of the tasks without the need of generating a previous task representation. Our clustering is dynamic and adapts to the new relation between gradients at each step of the gradients decent.
>
> 2. Lack of extensive experiments:
>  We agree with this point and will follow your recommendation.

---

### Official Review · AnonReviewer3 · 2020-10-28
**Official Blind Review #3**

**Rating:** 4
**Confidence:** 4

**Review:**

What is this paper about, what contributions does it make, what are the main strengths and weaknesses?
This paper proposed a learning algorithm of meta-learning: TreeMAML, to share information across tasks in meta-learning models. The paper compared the results of TreeMAML with MAML and the Baseline on SinusRegression Task and Linear Regression Task.

The main concerns are:

1.	Some statements in the introduction session are improper.

1.1	“there is still a lack of methods for sharing information across tasks in meta-learning models, and the goal of our work is to fill this gap.” See references:
a.	Invenio: Discovering hidden relationships between tasks/domains using structured meta learning. 2019.
b.	Hierarchical meta learning. 2019.
c.	Hierarchically structured meta-learning. 2019.
d.	Learning to propagate for graph meta-learning. 2019.

1.2	"However, these algorithms are not model-agnostic." Most of the related models are model-agnostic as long as replacing the task representation module.



2.	The contribution of this work is supposed to be emphasized. The innovation of this paper seems insufficient.

2.1	Yao et al. (2019). also aims to utilize the task relation and apply gradient-based adaptation methods. And the model structure presented in figure 1 is similar to the model framework in Yao’s paper.

2.2	As claimed in this paper, the hierarchical clustering algorithm, which is an essential part of the proposed model, is introduced with minor modification from Menon et al. (2019).

3.	As claimed in this paper, the proposed model lacks scalability since it works well only in the tree-structured tasks.

4.	The baselines exclude many relevant works such as the papers listed above.

---

> ### Author Response · Authors · 2020-11-23
> **Clarification about citations**
>
>  Thanks for your review:
>
>  We acknowledge in the paper the existence of other papers that try to use the data structure. We cite the work more related to us, including Yao's paper. We believe that the novelty of the approach is to modify the gradient descent to allow the NN to use the structure present on the data, our algorithm works in a completely different way than the one presented by Yao.

---

### Official Review · AnonReviewer1 · 2020-10-31
**While appropriately taking into account task similarity & structure in meta-learning and related fields is an important open problem, the scope of the paper is limited to a specific combination of existing algorithms on synthetic datasets. The submission also needs some work to make its methodology clearer.**

**Rating:** 3
**Confidence:** 5

**Review:**

The submission proposes a meta-learning algorithm attuned to the hierarchical structure of a dataset of tasks. Hierarchy is enforced in a set of synthetically-generated regression tasks via the data-sampling procedure, which is modified from the task-sampling procedure of [1] to include an additional source of randomness corresponding to which of a set of cluster components task parameters are generated from. The authors propose to adapt the model-agnostic meta-learning algorithm (MAML) of [1] to reflect this hierarchical structure by either observing (Section 4.1, FixedTree MAML) or inferring (Section 4.2, LearnedTree MAML) an assignment of tasks to clusters at each step of the inner loop (task-specific adaptation phase) of MAML; if tasks belong to the same cluster, the correspond task-parameters receive the same update at that step (in particular, the update direction is averaged). It is assumed that there are increasingly many clusters at each step, so that task-specific parameter updates are increasingly granular.

##### Strengths:
1) **Clarity**: The experimental setting and exactly how the data-generating process relates to the proposed algorithms are clearly described.
2) **Significance**: Results on the hierarchically structured synthetic regression task datasets demonstrate that {Fixed|Learned}Tree MAML: is at least as good as MAML, and often outperforms MAML; that it learns more efficiently than a MAML in terms of the cumulative number of datapoints observed; and that both MAML and {Fixed|Learned}Tree MAML outperform a naive baseline.

##### Weaknesses:
1) **Significance**: Since the evidence provided in favor of the proposed algorithm is in the form of an empirical evaluation on a synthetically generated dataset, the present impact of the algorithm is limited. In particular, there is no evidence that (i) the algorithm works for larger and/or more complex datasets; and (ii) that natural datasets of interest to the community exhibit a hierarchical structure analogous to the synthetic datasets presented in the submission.
2) **Novelty**: The algorithm modifies and combines previously introduced components: the MAML algorithm of [1]; the online top-down clustering algorithm of [2], and the task-similarity-as-gradient-similarity approach of [3].
3) **Clarity**: Specific details surrounding the relationship between Algorithm  1 and Algorithm 2 are insufficiently discussed:
  i) Algorithm 2 as it appears in the text is very similar to Algorithm 1 (The OTD algorithm) in [2] with the exception of the new hyperparameter $\xi$, and introduces new symbols that do not appear elsewhere in the text. It is therefore not sufficiently adapted for clarity in the context of this work.
  ii) Whether Algorithm 2 acts as a strict subroutine of Algorithm 2 is not stated. I believe it is not because the clustering decision for a new task relies on tree structures that are "generated for a training batch," although what a "training batch" refers to is not clear. Similarly, how the "online"/"offline" distinction in the context of the clustering algorithm fits into the training/evaluating setup borrowed from [1] is not made clear.
  iii) How exactly the task-similarity approach of [3] is employed in Algorithm 2 is not made clear. The only mention of the use of [3] is briefly around Eq. (8) before the main algorithm (Algorithm 1) is introduced, and Algorithm 2 only refers to a generic "similarity metric" (as in the original work, [1]).

##### References

[1] [Finn, Chelsea, Pieter Abbeel, and Sergey Levine. "Model-agnostic meta-learning for fast adaptation of deep networks." arXiv preprint arXiv:1703.03400 (2017).](https://arxiv.org/abs/1703.03400)

[2] [Menon, Aditya Krishna, Anand Rajagopalan, Baris Sumengen, Gui Citovsky, Qin Cao, and Sanjiv Kumar. "Online Hierarchical Clustering Approximations." arXiv preprint arXiv:1909.09667 (2019).](https://arxiv.org/abs/1909.09667)

[3] [Achille, Alessandro, Michael Lam, Rahul Tewari, Avinash Ravichandran, Subhransu Maji, Charless C. Fowlkes, Stefano Soatto, and Pietro Perona. "Task2vec: Task embedding for meta-learning." In Proceedings of the IEEE International Conference on Computer Vision, pp. 6430-6439. 2019.](https://arxiv.org/abs/1902.03545)

---

> ### Author Response · Authors · 2020-11-23
> **Comments about "Weaknesses section"**
>
> Thank you for taking the time to review this paper, and produce such a clear review. We provide clarifications to the "Cons" below:
>
> 1.We agree that the significance of the algorithm is more clear using real-world's dataset, and we will include this work in the future.
>
> 2.We believe that the Novelty of the approach is to modify the gradient descent to allow the NN to use the structure present on the data. This objective is already achieved with a fixed tree. Therefore the particular algorithm used for clustering is not fundamentally relevant, but use as an example of how to furder extend the uses of the method. Also, the fact that this paper is using MAML as a starting point does not speak against the novelty of the concept. MAML is the base of many meta-learning papers as REPTILE, foMAML, etc.
>
> 3.As stated above we did not try to introduce a new clustering algorithm in this paper and recognize that the used algorithm for clusterin was a simple modification of OTD. The clustering algorithm is run for each batch (following a batch the standard definition), generating a new tree each time in a dynamic fashion. In the synthetic examples, the similarity metric used is cosine similarity.

---

> > ### Comment · AnonReviewer1 · 2020-11-23
> > **Ideas have potential, but weaknesses cannot be addressed except in a future revision & resubmission**
> >
> > Re: novelty: I agree that the novelty of the approach lies in modifying MAML to capture the hierarchical structure. However, by modifying an existing algorithm, you necessarily reuse some of the ideas that were unique/novel to its original application. For example, in this case: The use of gradient descent to tune initial parameters in a meta-learning setup is attributable to MAML, not this work.
> >
> > As such, let me clarify my comment re: novelty: I evaluated the novelty attributable to the present submission to be small, exactly because it uses MAML as a starting point, but modifies the algorithm to be hierarchical in a particular sense and preliminarily applies it to corresponding hierarchical datasets. In particular, the novelty of the modification & application is the thing that I evaluated; I did not find this novelty to be significant in its present form, although the ideas have potential if my comment re: significance is addressed.
> >
> > Re: "The clustering algorithm is run for each batch (following a batch the standard definition), generating a new tree each time in a dynamic fashion."
> > This is an incomplete explanation that does not explain the training vs. new task distinction in the submission; e.g.:
> > "When a new task is evaluated, we reuse the tree structure that was generated for a training batch and add the new task. This saves us from computing a new task hierarchy from scratch for every new task."

---

### Decision · Program_Chairs · 2021-01-07
**Final Decision**

**Decision:**

Reject

**Comment:**

The paper proposes a variant of MAML for meta-learning on tasks with a hierarchical tree structure. The proposed algorithm is evaluated on synthetic datasets, and it compares favorably to MAML. The reviewers identified several significant weaknesses, including: (1) the experimental evaluation is limited, and it only includes small synthetic datasets; (2) the proposed algorithm is incremental over MAML. The reviewers agreed that the paper cannot be accepted in its current form. I recommend reject.